# High precision structured $H_\infty$ control of a piezoelectric nanopositioning platform

**Huan Feng[1], Hongbo Zhou[2], Congmei Jiang[1]\*, Aiping Pang[1]\***

**1** School of Electrical Engineering, Guizhou University, Guiyang, Guizhou Province, China, **2** School of Astronautics, Harbin Institute of Technology, Harbin, Heilongjiang Province, China

\* cmjiang3@gzu.edu.cn (CJ); appang@gzu.edu.cn (AP)

**Data Availability Statement:** All relevant data are within the paper.

**Funding:** This work was supported by the National Natural Science Foundation of China Regional Project 12162007, the Department of Education of

## Abstract

The inherent weakly damped resonant modes of the piezoelectric nanopositioning platform and the presence of model uncertainty seriously affect the performance of the system. A structured $H_\infty$ design is used in this paper to solve the accuracy and robustness problems respectively using a two-loop control structure. The multiple performance requirements of the system are constituted into an $H_\infty$ optimization matrix containing multi-dimensional performance diagonal decoupling outputs, and an inner damping controller $d$ is set according to the damping of the resonant modes; the second-order robust feedback controller is preset in the inner loop to improve the robustness of the system; the tracking controller is connected in series in the outer loop to achieve high accuracy scanning; finally, the structured $H_\infty$ controller is designed to meet the multiple performance requirements. To verify the effectiveness of the proposed structured $H_\infty$ control, simulation comparison experiments are done with the integral resonant control (IRC) and $H_\infty$ controller. The results demonstrate that the designed structured $H_\infty$ controller achieves higher tracking accuracy compared to the IRC and $H_\infty$ controllers under grating input signals of 5, 10, and 20 Hz. Moreover, it has good robustness under 600g and 1000g loads and high frequency disturbances close to the resonant frequency of the system, meeting multiple performance requirements. Compared with the traditional $H_\infty$ control, yet with lower complexity and transparency, which is more suitable for engineering practice applications.

## Introduction

With the emergence and development of nanotechnology, piezoelectric nanopositioning stages are widely used for precision motion control of many modern scientific instruments or industrial equipment, such as optical manipulation [1], biomedicine [2], micro-robotics [3, 4], MEMS [5]. The piezoelectric nanopositioning stage has the advantage of no electromagnetic interference, high resolution, fast response time, compact structure, no motion friction, without lubrication, etc. [6, 7], which can provide up to nanometer precision motion positioning for semiconductor processing, ultra-precision manufacturing, biofabrication, and nano-inspection. Recently, the expansion of applications around high-speed and high-precision manufacturing technologies requires higher and higher speed and positioning accuracy for

Guizhou Province, QianJiaoJi, China [2022]043 and the Science and Technology Fund of the Department of Science and Technology of Guizhou Province, Qian Kehe Foundation [2020] 1Y273. The funders had no role in study design, data collection and analysis, decision to publish, or preparation of the manuscript.

**Competing interests:** The authors have declared that no competing interests exist.

piezoelectric nanopositioning stages. However, the inherent low-damping vibration characteristics of the flexible mechanism inside the platform itself [8] and various types of uncertainties that exist in the actual system modeling and control, such as external disturbances, temperature variations, etc., limit the platform motion speed and positioning accuracy. Without proper treatment, the high-speed and high-precision motion positioning of the piezoelectric nanopositioning platform is hard to be realized. Therefore, to achieve high speed and high accuracy motion positioning of piezoelectric nanopositioning platform. The weakly damped vibration characteristics of the system and the robustness problem with model uncertainties must be solved.

To explore the performance limits of nanopositioning platforms with respect to speed and accuracy, the damping control and tracking control methods have been proposed and widely investigated. The feed-forward damping architecture-based control methods have been proposed to compensate the resonant vibrations caused by weakly damped modes, such as inversion-based compensators [9], model-free iterative learning control (ILC) [10], and ILC with linear time-varying (LTV) Q filters [11], however the drawback of feed-forward-based control methods is the low robustness of the system when model uncertainty exists. Subsequently, the feedback control architectures which suppress weakly damped resonant modes and improve tracking performance have been extensively investigated. The high-bandwidth damped controllers are designed to solve the system resonant vibration problems, such as robust $H_\infty$ control [12], μ-synthesis [13], linear quadratic Gaussian (LQG) regulator [14], and sliding mode control. Nonetheless, these kinds of controllers often depend on the order of the system. For high-order systems, design processes are often complex, moreover, the engineering cost of such controllers is high as well as their implementation is difficult. Thus, control methods based on negative imaginary number (NI) theory have been proposed. Negative imaginary number feedback damping controllers [15] were investigated for suppressing resonant modes with flexible structures such as piezoelectric nanopositioners. For NI systems, a rigorous NI controller can be designed to ensure the internal stability of the positive feedback interconnection of the two systems. Such as positive position feedback (PPF) [16], positive velocity and position feedback (PVPF) [17], positive acceleration, velocity, and position feedback (PAVPF) [18], integral force feedback (IFF) [19], passive parallel damping (PSD) [20], resonant control (RC) [21], recursive delayed position feedback (RDPF) [22], and integral resonant control (IRC) [23]. These controllers are fixed in structure, low order and easy to implement. Among the above studies, IRC is a simple, low-order control method for suppressing the vibration of a multiresonant mode system while maintaining a large stability margin. But the selection and adjustment of its controller parameters do not take into account the model uncertainty caused by load changes or changes in the surrounding environment, whose robustness needs to be improved. In addition, other controllers have some problems in the application. For instance, the PPF controller cannot realize the arbitrary configuration of the second-order poles in the s-plane; the integral force feedback control (IFF) requires the addition of force sensors, etc. Apart from feedforward and feedback controllers, combined feedforward/feedback controllers are another commonly used hysteresis loop control method that has been discussed in many papers [24]. The input shaping method can also suppress mechanical vibrations by reducing the amplitude of high-frequency components near or above resonance [25]. An integral force feedback hybrid control system with input shaping was developed by combining the integral force feedback with the input shaping method [26], which reduces the effect of hysteresis nonlinearity and suppresses the vibration of the closed-loop system. The above studies mainly focus on whether the resonant modes are suppressed to improve stability, or whether the structure is simple and improves tracking accuracy, without comprehensive requirements on the performance of the system in terms of stability, robustness, bandwidth, and accuracy.

$H_\infty$ control is a comprehensive control theory that can integrate various design requirements of the system, such as robust stability, system bandwidth requirements, output performance, etc. The literature [27] gives an $H_\infty$ controller that enhances the robustness of the system while achieving high accuracy, but the limitation is that the proposed $H_\infty$ controller is high order and complex. Furthermore, in engineering practice, it requires empirical decomposition of the complex controller into multiple low-complexity control structures for implementation, which is a great challenge and difficult to achieve in practical applications.

Recently, Apkarian et al [28, 29] proposed a new structured $H_\infty$ integrated control method. The core of structured $H_\infty$ control theory is a locally optimal control strategy that balances control performance and controller complexity. Firstly, the reasonable controller structure is designed according to the system characteristics and control objectives. Secondly, the "standard H∞ parametric matrix" with multiple performance decoupled outputs and multiple controllers mixed and nested is derived to build a standard structured integrated control form. Finally, the reasonable weighting function is designed to solve for the optimal controller parameters. Compared with the traditional $H_\infty$ control method, the advantage of the structured $H_\infty$ control design method is that the order or structure of the controller can be predetermined, which finally results in a controller that can meet the system performance requirements and relatively simple.

This paper adopts a structured $H_\infty$ control method to pre-set the controller structure for the inherent weakly damped resonant modes, model uncertainty and high-precision scan tracking problems of the piezoelectric-driven nanopositioning platform. An internal damping controller $d$ (whose parameters are fixed and non-adjustable) is set according to the damping of the resonant modes, aiming to add a corresponding weakly damped zero point to be the fast return of the weakly damped poles of the system. In addition, an internal loop robust feedback controller is preset to push the dominant pole of the system away from the negative imaginary axis direction to improve the robustness of the system. Under the premise that the inner loop satisfies the robustness constraint, the outer loop is connected in series with a PI (proportional-integral) controller to achieve high-precision scan tracking. By analyzing the system requirements and selecting the appropriate performance weighting function according to the control objectives to ensure each performance requirement, the performance decoupling output matrix with parameters is constructed. Finally, the performance weighting matrix constructed above is optimized parametrically, the structured $H_\infty$ parametrization is calculated, and the optimal controller parameters satisfying the multi-performance requirements are solved. The simulation experiments show that the designed structured controller can meet multiple performance requirements with simple structure and easy implementation.

## Model

The piezoelectric nanopositioning stage is a precision motion system driven by smart materials, which makes it a complex mechatronic system due to its system composition including a piezoelectric ceramic actuator, a voltage amplifier and a nanopositioning stage based on a flexure mechanism. Also, since the nano-motion system assumes an extremely important role in the fields of micro and nano inspection and manufacturing, and ultra-precision manipulation, it requires precise control to achieve the performance requirements of the motion system in terms of positioning accuracy, resolution, transient response and anti-interference capability. Consequently, the modeling of the nanopositioning platform system, as the basis of motion control, is of particular importance. However, for hysteresis models in piezoelectric positioning systems, there are physics-based modeling [30] and phenomenological models [31]. Physics-based models are usually very complex, and a physics-based model developed for one

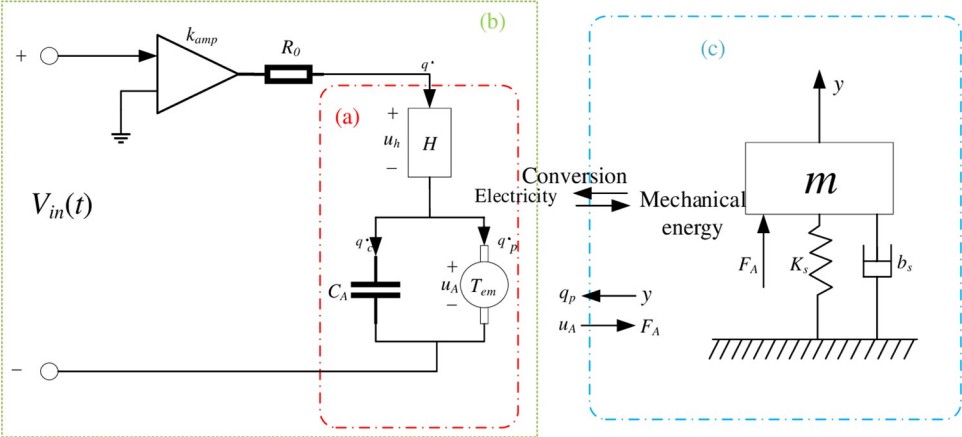

**Fig 1. Schematic diagram of the piezoelectric positioning system.**

material may not be applicable to another material. In contrast, phenomenological models are used to produce behavior similar to that of physics-based models. Depending on the mathematical structure, the phenomenological models describing the hysteresis phenomenon of piezoelectric actuators can be classified into hysteresis models based on differential equations, hysteresis models based on operators, and other hysteresis models. The common differential equation-based models are the Duhem model [32], the Backlash-like model [33], and the Bouc-Wen model [34]. However, differential equation-based models are difficult to obtain general solutions, on the other hand, there is no method to construct an analytical inverse of a differential equation-based hysteresis compensation model so far. The identification and implementation of Preisach models, the most typical of operator-based hysteresis models [35], is usually very complicated due to the presence of double integrals. In addition, the Preisach model is not analytically invertible, which generally uses numerical methods to find the approximate inversion of the model in practice. Besides these two models, other models have been used to predict lags, such as intelligent models based on artificial neural networks [36], fuzzy systems [37], and support vector machines [38]. It is worth noting that these hysteresis models can accurately and effectively describe the hysteresis of piezoelectric drives in specific applications, but so far there is no general model to fully represent the hysteresis behavior of the piezoelectric drive stage, so the establishment of a hysteresis model for the system is a major difficulty problem. This paper combines the literature [39] to construct a relatively accurate comprehensive dynamics model of the piezoelectric positioning platform, Fig 1 shows the schematic diagram of the piezoelectric positioning system.

Fig 1(A) shows the piezoelectric ceramic driver model, which divides the hysteresis and piezoelectric effects into two relatively independent sub-models, where $H$ represents the effect of hysteresis of the ceramic material, and $u_h$ is the voltage caused by the hysteresis effect; $T_{em}$ represents the piezoelectric effect, $u_A$ is the voltage generated by the piezoelectric effect, with $q_p$ being the charge generated by the piezoelectric effect; $C_A$ represents the total capacitance of all piezoelectric ceramic wafers connected in parallel, while $q_c$ is the charge stored in the capacitor $C_A$. The charge of the whole circuit is denoted as $q$, with the current flowing through the whole circuit denoted as $\dot{q}$; $y$ denotes the actual elongation of the driver.

$$u_h(t) = H(q) \tag{1}$$

$$q(t) = q_c(t) + q_p(t) \tag{2}$$

$$u_A(t) = q_c(t)/C_A \tag{3}$$

$$q_p(t) = T_{em}y(t) \tag{4}$$

Consider the actual system using an amplifier circuit to generate the drive voltage or charge to make the piezoelectric ceramic driver act. Here, while the charge driver can be used to achieve the suppression of the hysteresis nonlinear effect of the driver, but due to the complexity and high cost of the charge amplifier design, a voltage amplifier is used to excite the piezoelectric ceramic driver in the actual system design, where the amplifier circuit model is Fig 1 (B). In which, $R_0$ is the equivalent internal resistance of the driver amplifier circuit, $k_{amp}$ represents the amplification factor of the voltage amplifier, and $V_{in}(t)$ is the total voltage applied to the driver. Based on the Kirchhoff voltage effect of the circuit, it is obtained that:

$$R_0\dot{q}(t) + u_h(t) + u_A(t) = k_{amp}V_{in}(t) \tag{5}$$

In view of the motion relationship between the piezoelectric ceramic actuator and the flexible amplification mechanism in the piezoelectric nanopositioning platform and the active body of the platform, the mechanical transmission relationship of the piezoelectric nanopositioning platform can be simplified into a mass-damped-spring second-order system, as shown in the equivalent mechanical dynamics model in Fig 1(C). The equivalent mechanical model is resolved according to Newton's laws of motion and piezoelectric effects as follows.

$$F_A = T_{em}u_A(t) \tag{6}$$

$$m\ddot{y}(t) + b_s\dot{y}(t) + k_s y(t) = F_A \tag{7}$$

where $m$, $b_s$ and $k_s$ indicate the total mass, damping coefficient and overall stiffness of the active body of the piezoelectric nanopositioning platform, respectively; $F_A$ is the mechanical thrust generated by the piezoelectric ceramic actuator. When the voltage is loaded on the piezoelectric ceramic, the piezoelectric ceramic elongates because of the piezoelectric effect and generates the thrust $F_A$ in the contact and interaction with the platform, which drives the piezoelectric positioning platform to produce the output displacement $y$. By associating Eq (1) ~ (7), the overall dynamics of the nanopositioning platform can be modeled as follows,

$$\dddot{y}(t) + a_2\ddot{y}(t) + a_1\dot{y}(t) + a_0(t) = b_1 V_{in}(t) - b_0 H(q) \tag{8}$$

The parameters in the formula are expressed as follows,

$$b_0 = \frac{T_{em}}{mR_0 C_A}$$

$$b_1 = \frac{T_{em}k_{amp}}{mR_0 C_A}$$

$$a_0 = \frac{R_0 k_s C_A + (R_0 - 1)T_{em}^2}{mR_0 C_A}$$

$$a_1 = \frac{k_s C_A + T_{em}^2 + b_s}{mC_A}$$

$$a_2 = \frac{b_s C_A + m}{mC_A}$$

For the avoidance of nonlinear effects caused by voltage hysteresis, a low-amplitude input voltage (a sinusoidal scanning input voltage with a constant amplitude of 200 mV between 0.1 Hz and 500 Hz applied to the *y*-axis) was used in the literature [40] to avoid hysteresis-induced distortion [41]. Using the MATLAB identification toolbox, the piezoelectric nanopositioning platform with weakly damped modes is obtained without mechanical loading dynamics model shown in Eq (9) whose identification details can be found in the literature [40].

$$G(s) = \frac{y(s)}{u(s)} = \frac{1.196 \times 10^6}{s^2 + 110s + 1.673 \times 10^6} \tag{9}$$

where *Y(s)* and *U(s)* are the output displacement and the input drive voltage, respectively. In the actual modeling and control process, there will be various uncertainties, among which the most influential is the model uncertainty caused by different mechanical loads. The frequency response of the system identification model obtained when the mechanical loads are 0g (nominal system), 600g and 1000g respectively is shown in Fig 2.

The inherent resonant mode of the nominal system appears at 205 Hz and the resonance amplitude is 18.5dB. The resonant modes of the system at mechanical loads of 600 g and 1000 g appear at 171 Hz and 139 Hz with resonance amplitudes of 8.72 dB and 3.48 dB, respectively. While suppressing the inherent resonant mode of the system, the model uncertainty caused by the mechanical load variation cannot be ignored, if not properly handled, system control accuracy will be seriously affected. Thus, there is an urgent need to design a controller that meets the requirements of suppressing the inherent resonant modes of the system and ensuring the robustness of the system, so as to realize the task of high-speed and high-precision scanning and tracking of the platform.

## Structured $H_\infty$ control design

The traditional $H_\infty$ design is based on the theory of singular value decomposition of matrices. With the maximum singular value of the transfer function matrix as a measure of a comprehensive optimal control design, the control problem containing multiple performance requirements is reduced to the $H_\infty$ standardized design problem of solving the optimal matrix parametric index of the weighted comprehensive performance. Although $H_\infty$ control has many advantages, there are also many application limitations, mainly that such designs have

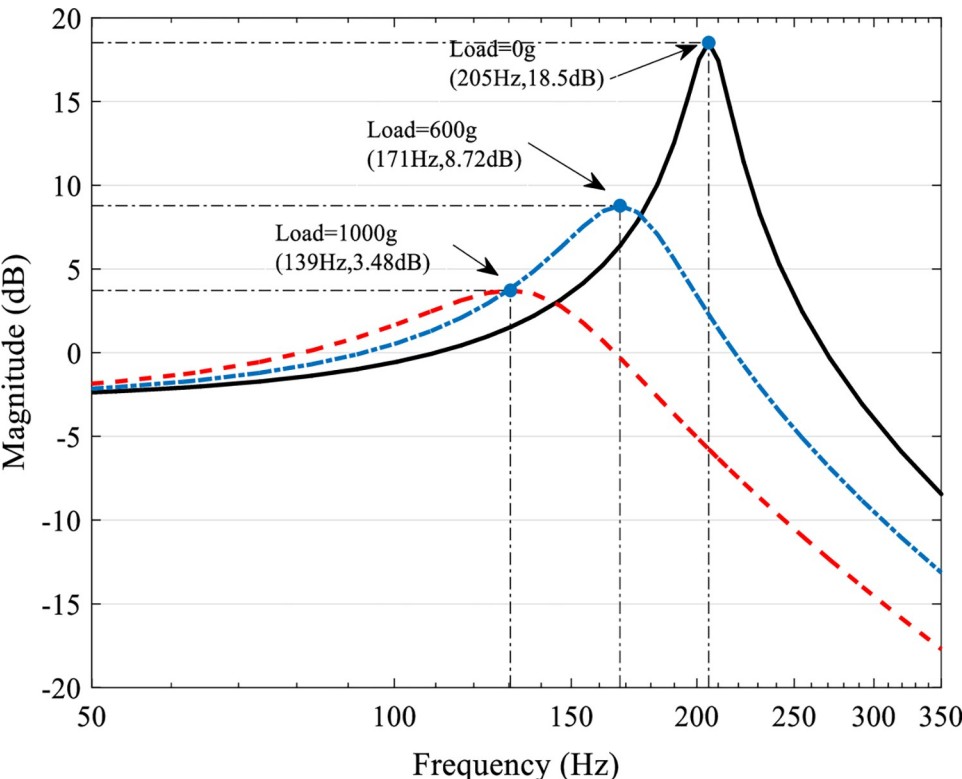

**Fig 2. Open-loop bode diagram of the system at mechanical loads of 0g, 600g and 1000g.**

only one parametric metric and multiple performances are coupled with each other. The design is highly opaque because it is not easy to analyze each performance due to the constraints of a single parametric indicator. In addition, the traditional $H_\infty$ control design only seeks to achieve the optimal comprehensive performance, the complexity of the controller is not limited, the most available controller is often of high order and complex structure, not suitable for practical engineering applications.

In this paper, a new structured $H_\infty$ control scheme is proposed. Compared with the traditional $H_\infty$ design method, its superiority lies in the fact that the order and structure of the controller can be predefined to give a relatively low-order structured $H_\infty$ controller. It retains the comprehensiveness of the traditional $H_\infty$ design, but also allows a weighted design for each performance requirement, constituting a diagonal matrix containing multidimensional performance outputs for comprehensive performance optimization.

## Controller structure design

Taking into account the accuracy and robustness requirements of the piezoelectric-driven nanopositioning platform, a pre-defined structured $H_\infty$ controller structure is given in Fig 3. This control structure contains two loops, an inner-loop positive feedback loop for damping control and an outer-loop negative feedback loop for improving the tracking accuracy. Where $G$ is the nominal model, $d$ is the internal damping control term $C_d$ is the robust controller, and $C_t$ is the tracking controller. $u$ is the tracking control input, $u_d$ is the control input perturbation, $y_i$ and $y$ are the reference input and system displacement output, respectively, and $\Delta$ is the multiplicative uptake of the object. The red dashed box $C(s)$ is the optimizable part of the structured design, and $d$ is based on the actual object design.

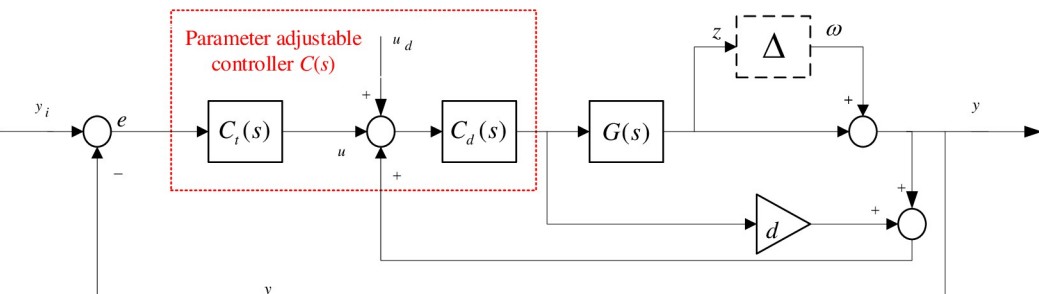

**Fig 3. Block diagram of the control structure of the system.**

The design ideas of each of these control structures are as follows,

1): Feed-through items

The dynamics of the piezoelectric nanopositioning table with weakly damped modes can be modeled as a second-order system that,

$$G(s) = \frac{y(s)}{u(s)} = \frac{\sigma^2}{s^2 + 2\zeta_n \omega_n s + \omega_n^2} \tag{10}$$

where $\sigma^2$ is the low-frequency gain of the system, $s$ is the Laplace operator of the continuous system, and $\zeta_n$ and $\omega_n$ are the damping coefficient and the intrinsic frequency of the system, respectively. In this system, $\zeta_n \ll 1$, which indicates a weakly damped resonant mode at $\omega_n$.

The purpose of adding the feedthrough term $d$ is to produce a pair of zeros $z_{1,2} = \pm j\omega_z$ in the root trajectory of the damping loop, satisfying $\frac{\omega_n}{3} < \omega_z < \omega_n$. As the controller gain increases, the root trajectory will start at the natural pole and end at the increased resonance zeros. This is well explained and analyzed in [23]. The analytical analysis yields that the feed-through term is chosen as $d = \frac{-2\sigma^2}{\omega_n^2} = -1.43$

2): Robust controller

To make the controller robust to model uncertainties and external perturbations and enhance the immunity of the system to disturbances. Considering the need to implement a low-order controller to meet the performance requirements, a second-order robust controller of the following form is designed in the inner loop after analysis.

$$C_d(s) = \frac{a_0 s^2 + a_1 s + a_2}{s^2 + a_3 s + a_4} \tag{11}$$

where $a_i(i = 0,1,2,3,4)$ is the parameter to be optimized.

3): Tracking controller

The classical tracking controller is generally in the form of integration, but the system in this example has high accuracy requirements. In order to further improve the accuracy and response speed of the system, minimize the steady-state error and achieve high precision high-speed scanning tracking. In this paper, the PI tracking controller is designed as follows.

$$C_t = k_p + \frac{k_i}{s} \tag{12}$$

where $k_p$ and $k_i$ are the parameters to be optimized.

## Performance weighted design

Fig 2 shows the open-loop amplitude-frequency characteristics of the system under mechanical loads of 0, 600 and 1000 g, respectively. It can be seen from the figure that the nominal system has an inherent low damping resonant mode at 205 Hz, whose resonance peak reaches 18.5 dB. Other mechanical loads also have certain resonant modes at the corresponding frequencies. When the external disturbance has a high frequency component close to the resonant frequency of the platform, it will excite the resonant vibration of the system, and the serious resonance will even damage the system hardware. Therefore, the first performance requirement of the system is to suppress external disturbances.

The model uncertainty of the system is induced when the surrounding environment of the system changes or scanning items with different mechanical loads at high speed, which can be considered as an unmodeled dynamics module (model uncertainty of the system). In studying the control problem, only the ontology module of the piezoelectric nanopositioning platform is modeled, while all others are considered as uncertainties of the system. The uncertainty of the system is expressed in terms of multiplicative uncertainty:

$$G_a(s) = G(s)[1 + \Delta(s)] \tag{13}$$

Where $\Delta$ is the multiplicative uptake, $G_a$ is the real system model under uptake, and $G$ is the nominal model of the system without uptake. The block diagram of the system except for the ingestion parameter $\Delta$ is reduced and combined to obtain the transfer function $T_{z\omega}$ from $\omega$ to $z$. The equivalent block diagram is shown in Fig 4.

According to the small gain principle, the system with robust stability needs to satisfy the condition of Eq (14):

$$\|T_{z\omega}\|_\infty < 1 \tag{14}$$

Thus, the second performance requirement of the system is robust stability.

In addition, while ensuring that the system has robust stability, the bandwidth of the control system should be increased to achieve high precision and high speed scanning and tracking of the excitation signal, so the third performance requirement of the system is bandwidth.

Following the analysis above, the performance requirements for this control system design are suppression of external disturbances, bandwidth and robust stability requirements. In the

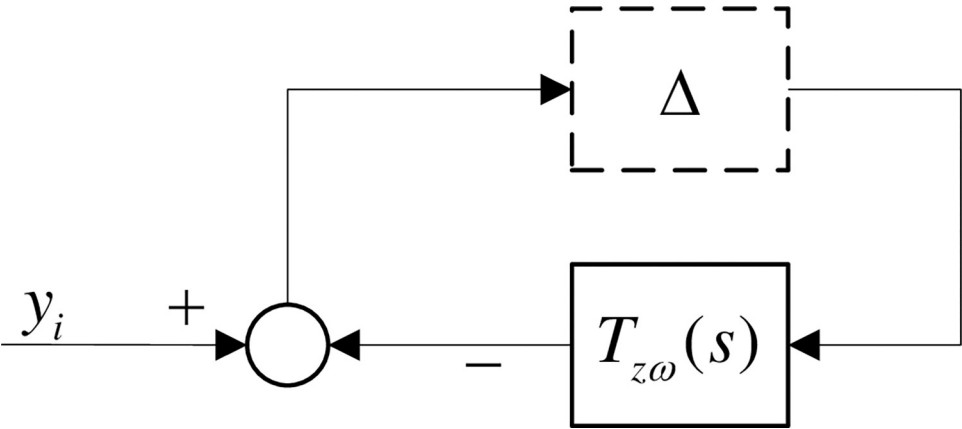

**Fig 4. Equivalent block diagram of control structure.**

following, the appropriate weighting functions are selected for each of the above three system performance requirements and then the controller is solved.

Let $T_1$ be the transfer function from $y_i$ to $e$. The stability of the system, that is, the distance of the open-loop transfer function to the critical stability point, which is also the upper limit of the gain of the sensitivity function, requires that

$$\|W_1(s)T_1(s)\|_\infty \le \gamma \tag{15}$$

Where $W_1(s)$ is the weighting function and $\gamma$ is the parametric index. In order to meet the stability of the system and ensure that the system can achieve high precision tracking requires that the controller contains an integration link, so the integration law is required in $W_1(s)$, that is

$$W_1(s) = \frac{\rho}{s + 0.001} \tag{16}$$

The 0.001 is an additional regenerative quantity to avoid the appearance of the pole on the imaginary axis, where $\rho$ is the parameter to be selected in the $H_\infty$ design.

Let $T_2$ be a transfer function from $\omega$ to $z$. Considering the possible uncertainties in the system caused by various un-modeled dynamics, for robust stability of the system requires that

$$\|W_2(s)T_2(s)\|_\infty \le \gamma \tag{17}$$

To ensure the robustness of the system under different mechanical loads, the multiplicative uncertainty $\Delta$, which can be calculated under 600g and 1000g loads by Eq (13) after obtaining the nominal and regressive models of the system, as shown in Fig 5. For all uncertainties, the suitable weighting function $W_2$ is designed, and for all frequency bands, so that the

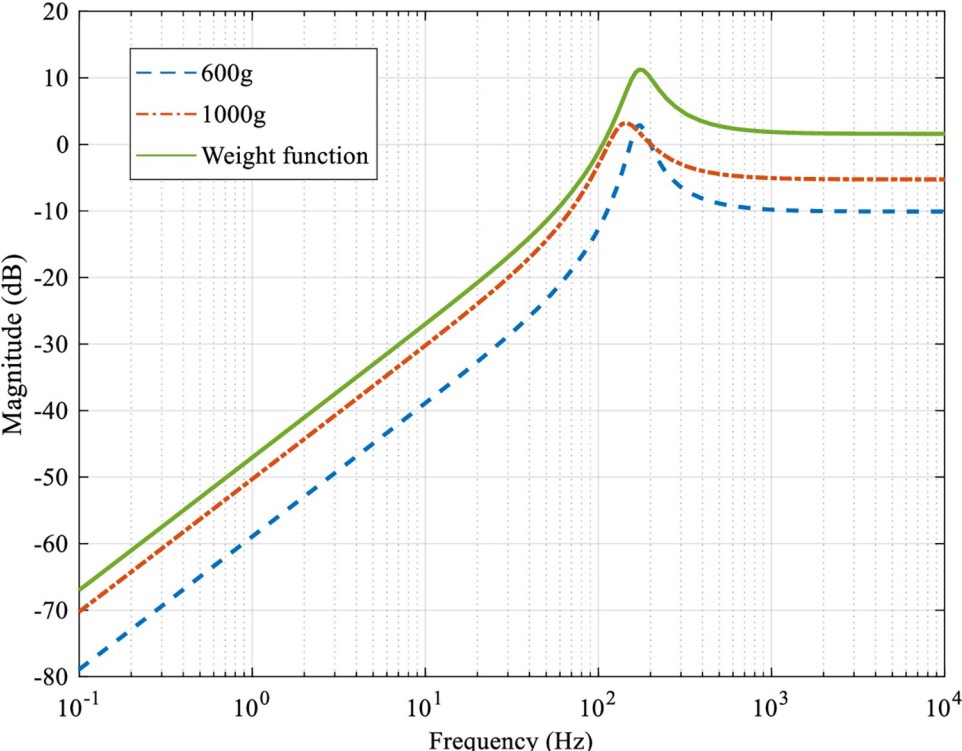

**Fig 5. Weighting function and multiplicative regression diagram under load.**

multiplicative uncertainty $\Delta$ needs to satisfy $\|\Delta\| \leq \Delta_m \cdot W_2$, where $\Delta_m$ is the unitized regression. The weighting function $W_2$ designed for the experiments in this paper is chosen as

$$W_2 = \frac{1.2(s + 0.08)(s + 676)}{s^2 + 419s + 1.15 \times 10^6} \tag{18}$$

Let $T_3$ be the transfer function of $y_i$ to $y$. The bandwidth requirement of the system is

$$\|W_3(s)T_3(s)\|_\infty \leq \gamma \tag{19}$$

The weighting function $W_3(s)$ is chosen to be a limit to the bandwidth of the system, while requiring the closed-loop characteristics after the bandwidth to -40dB/dec decay, which is taken as $W_3(s) = \frac{s(0.00005s+1)}{10000}$ in this paper. In addition, the general power functions in the $H_\infty$ solution process are true rational functions, so a small time constant is added to $W_3(s)$ to ensure that the numerator denominator is of the same order. Thus, the final robust stability weighting function $W_3(s)$ can be chosen as

$$W_3(s) = \frac{s(0.00005s + 1)}{10000(0.00001s + 1)^2} \tag{20}$$

## Structured $H_\infty$ controller

In view of the above analysis, the control performance requirements of the piezoelectric nano-positioning system are considered comprehensively for the designed controller structure and the structured $H_\infty$ optimization problem of the weighting function. By optimizing the adjustable parameters $k_p$, $k_i$ and $a_i (i = 0,1,2,3,4)$, the minimum $\gamma$ value satisfying Eq (21) is obtained.

$$\|H\|_\infty \leq \gamma \tag{21}$$

in which $H = diag(W_1 T_1\ W_2 T_2\ W_3 T_3)$. The adjustable parameters obtained at this point are the optimal parameters for the system controller.

When finding the optimal parameters in the structured $H_\infty$ controller, $T_1$, $T_2$, $T_3$ of Eq (21) are expressed in the following linear fractional form, applying the linear fractional transform (LFT) method [28, 29] to extract the parameters in the structured controller for parameter optimization design.

$$\begin{cases} T_1 = \mathrm{F}_l(P_1, C) \\ T_2 = \mathrm{F}_l(P_2, C) \\ T_3 = \mathrm{F}_l(P_3, C) \end{cases} \tag{22}$$

When $\rho = 25000$ in the performance weight function $W_1$, the optimal metric of the parameters $\gamma = 2.03$ is obtained. The corresponding optimal parameters of the structured controller

$$a_0 = 0.0012, a_1 = 1.2853, a_2 = 3.7795 \times 10^3,$$

are obtained as follows. $a_3 = 2.6227 \times 10^4, a_4 = 5.3627 \times 10^4,$

$$k_p = 1.1979 \times 10^5, k_i = 2.5612 \times 10^5$$

## Control performance analysis

Fig 6 represents the relationship between $T_1$ and $\frac{\gamma}{W_1}$ after solving for the optimized parameters of the controller, which can be seen to satisfy the requirements of Eq (15) for sensitivity, and the right function $W_1(s)$ is selected appropriately.

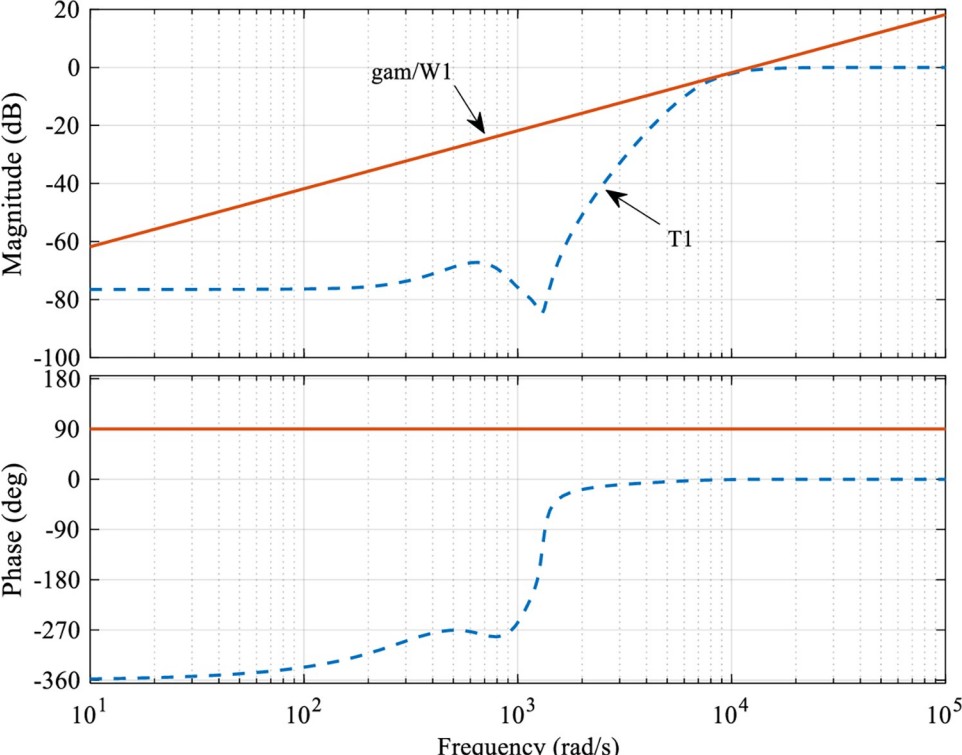

**Fig 6. Bode plot of the singular values of the sensitivity function $T_1$.**

The relationship between the transfer function $T_2$ and $\frac{\gamma}{W_2}$ after obtaining the optimal parameters of the controller is shown in Fig 7. As can be seen, the selection of the power function $W_2$ satisfies Eq (17) and meets the robust stability requirements. Moreover, the amplitude-frequency characteristics of $T_2$ are quite far from $\frac{\gamma}{W_2}$, which indicates that it is extremely robust to the uptake of the control object.

The relationship between the transfer function $T_3$ and $\frac{\gamma}{W_3}$ after obtaining the optimal parameters of the controller is shown in Fig 8, which shows that the selection of the weight function $W_3$ satisfies Eq (19) and meets the set bandwidth requirements.

## Simulation comparative analysis

This section addresses the issues of high precision tracking, robustness and interference immunity of the piezoelectric driven nanopositioning platform. For verifying the effectiveness of the controllers designed in the paper, simulation analysis is performed to compare with the classical IRC control and $H_\infty$ control, respectively. Among them, the design results of IRC and $H_\infty$ controllers are as follows.

Exploiting the IRC parameter resolution method introduced in the literature [23], the IRC in this example is designed as:

$$d = -1.43, \; C_d^{IRC} = \frac{980}{s}, \; C_t^{IRC} = \frac{430}{s}$$

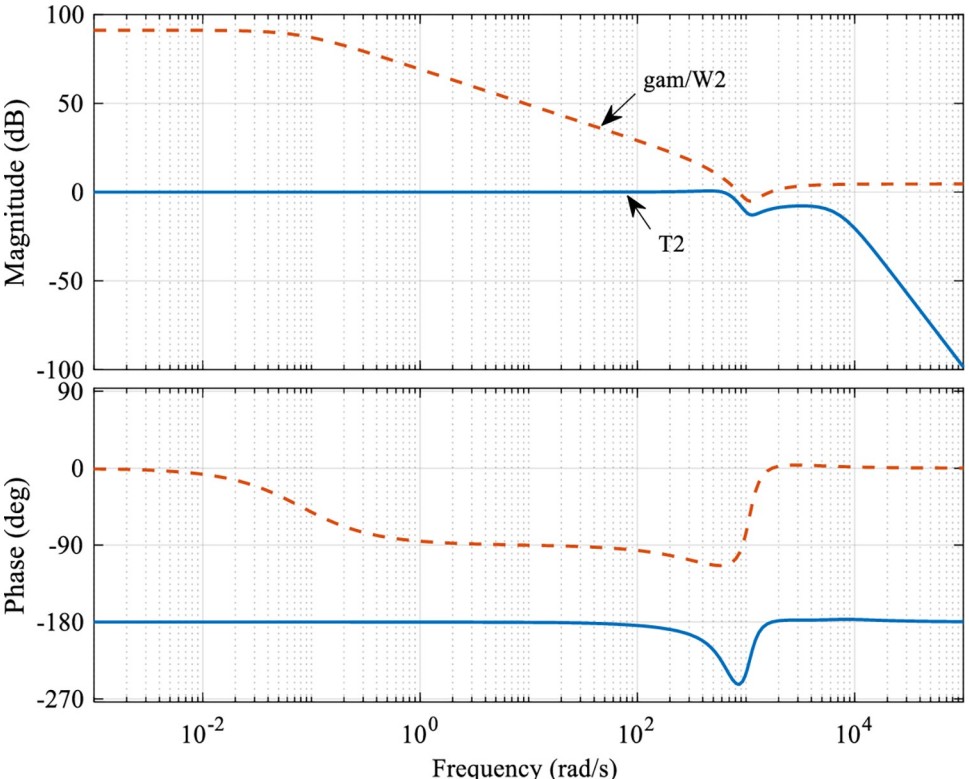

**Fig 7. Singular value bode diagram of the closed-loop transfer function $T_2$.**

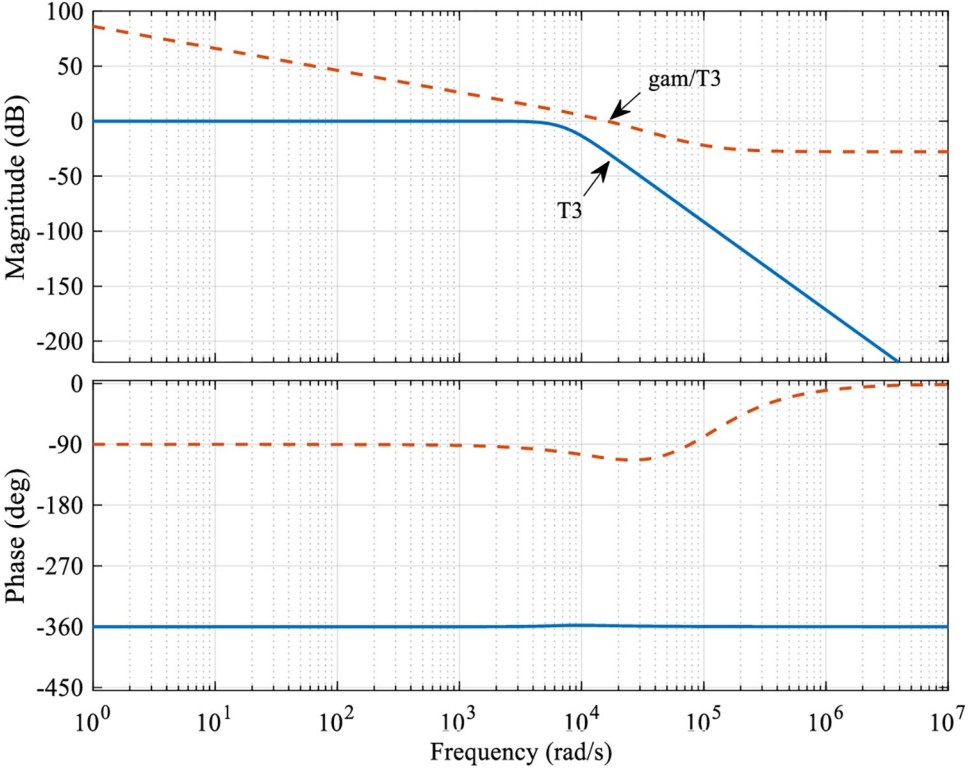

**Fig 8. Singular value bode diagram of the closed-loop transfer function $T_3$.**

The $H_\infty$ control method proposed in the literature [27] for the piezoelectric nanopositioning platform is used, and its controller is designed as follows:

$$K_\infty(s) = \frac{-5.208 \times 10^9 (s+1220)(s+1 \times 10^5)(s^2+1754s+9.226 \times 10^5)(s^2+75.03s+7.629 \times 10^5)}{(s+6.024 \times 10^7)(s+0.01)(s^2+2374s+1.374 \times 10^6)(s^2+1746s+7.621 \times 10^5)(s^2+1.046 \times 10^6 s+5.473 \times 10^{11})}$$

For systems under different loads (including 0g, 600g and 1000g), grating signals with different frequencies are set as reference inputs (frequencies of 5, 10 and 20Hz and amplitudes of $2\mu m$), while high frequency interference close to the resonant frequency of the system is considered. The comprehensive superiority of the accuracy, robustness, and interference immunity of the designed structured $H_\infty$ controller is verified.

## Grating signal tracking test

For evaluating the tracking performance of the three controllers, a group of grating scan signals at 5, 10 and 20 Hz are input to the platform. IRC, $H_\infty$ controller, structured $H_\infty$ controller without high frequency disturbance signal for different frequencies of the grating input signal tracking results are shown in Fig 9, and the root mean square error of tracking (RMSE) results are shown in Fig 10. Fig 9 reflects that all three controllers track the grating reference input signals at 5 Hz and 10 Hz well under 0 g mechanical load. Among them, in the case of 5 Hz, the RMSEs of the IRC and $H_\infty$ controllers are 0.04637 $\mu m$ and 0.02143 $\mu m$, respectively, while the RMSE of the structured $H_\infty$ controller is 0.001618 $\mu m$, which improves 96.5% and 92.4% over the IRC and $H_\infty$, respectively. As the frequency increases to 20 Hz, the tracking becomes worse under the IRC controller, with an RMSE value of 0.1859 $\mu m$, while slightly worse under the $H_\infty$ controller, with an RMSE value of 0.08391 $\mu m$. However, high accuracy tracking is always maintained under the structured $H_\infty$ controller, with a root mean square error (RMSE) value of 0.00647 $\mu m$, which is one order of magnitude smaller than the error under the previous two controllers.

## Robustness test

Since the system has model uncertainty, the first resonant frequency of the system shifts from 205 Hz to 139 Hz under a mechanical load of 1000g. So to verify the robust performance of the three controllers under model uncertainty, a grating scan signal with the same frequency is input for the system under different mechanical loads. Experimental simulations of IRC, $H_\infty$ controller and structured $H_\infty$ controller were compared and analyzed, and the root mean square error (RMSE) of the system output displacement under the three controllers are shown in Table 1.

It can be seen that all three controllers have good robustness under different mechanical loads. But the tracking accuracy is higher with the structured $H_\infty$ controller. The tracking results of the system under the most demanding conditions (mechanical load of 1000g and input grating signal of 20Hz) are shown in Fig 11, which shows that the structured $H_\infty$ controller has good robustness and high accuracy tracking. That is because the structured $H_\infty$ controller improves the closed-loop bandwidth of the system while ensuring the robust stability of the system. Under a mechanical load of 0g, the closed-loop bandwidth of the IRC controller is 173Hz, the closed-loop bandwidth of the $H_\infty$ controller is 106Hz, and the bandwidth of the structured $H_\infty$ controller is 1445Hz, resulting in a faster response time and higher accuracy, as demonstrated in the grating signal tracking experiments. In addition, at a mechanical load of 1000g, the closed-loop bandwidth of the system under the structured $H_\infty$ controller is 573Hz, which is still larger than the closed-loop bandwidth of the previous two controllers, which still achieves high accuracy and high speed tracking within the bandwidth.

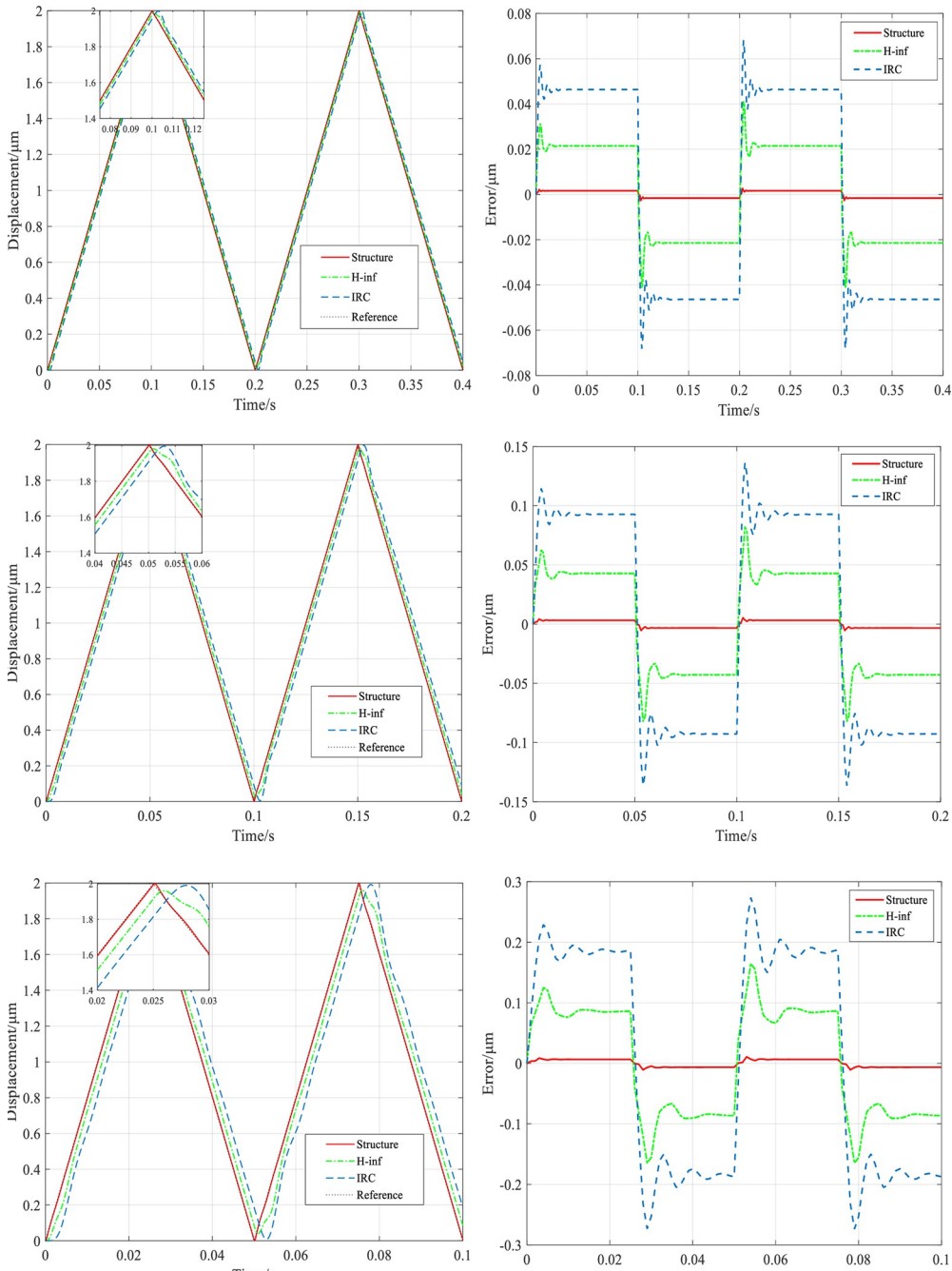

**Fig 9. System tracking results for 5, 10 and 20Hz grating input signals at 0g mechanical load.** (a) Tracking 5Hz grating signal output and tracking error experimental results. (b) Tracking 10Hz grating signal output and tracking error experimental results. (c) Tracking 20Hz grating signal output and tracking error experimental results.

## High frequency signal disturbance performance test

To verify the anti-interference performance of the designed controller, a high frequency sinusoidal interference signal $u_d = 0.2\sin(139t)$ which is close to the resonant frequency of the system (139Hz) was added to the system at a mechanical load of 1000g with an input frequency of

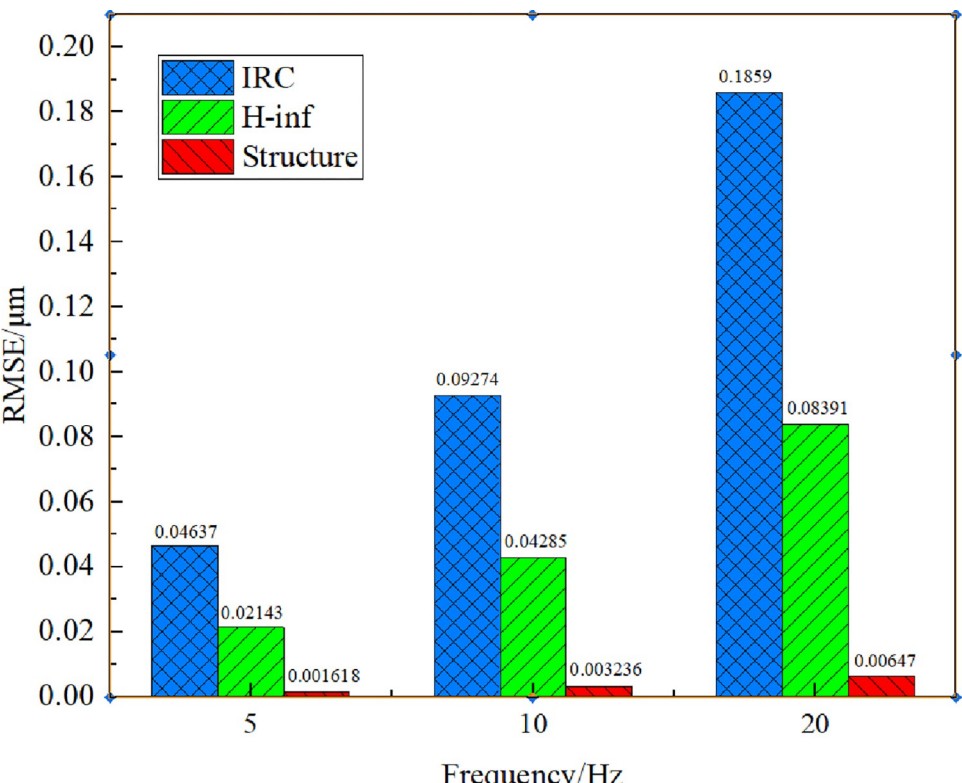

**Fig 10. System tracking RMSE for 5, 10 and 20Hz grating signals at 0g mechanical load.**

20Hz, whose tracking error results are shown in Fig 12. The tracking RMSE values of the system under IRC, $H_\infty$ controller and structured $H_\infty$ controller are 0.1926$\mu m$, 0.2463$\mu m$ and 0.01174$\mu m$, respectively, and the RMSE under structured $H_\infty$ controller is 93% and 95% less than that under IRC and $H_\infty$ controller, respectively. The tracking error results are shown in Fig 13 with the addition of a high frequency sinusoidal interference signal $u_d = 0.2\sin(205t)$ close to the resonant frequency (205 Hz) of the system at this time under a mechanical load of 0g. The tracking RMSE values of the system under IRC, $H_\infty$ controller and structured $H_\infty$ controller are 0.2477$\mu m$, 0.2252$\mu m$ and 0.01175$\mu m$, respectively. The RMSE under the structured $H_\infty$ controller is less than IRC and $H_\infty$ controller by 95.2% and 94.8%, respectively.

**Table 1. Grating tracking experimental performance test results.**

| Reference Frequency(Hz) | Load(g) | RMSE With IRC($\mu m$) | RMSE With H-inf($\mu m$) | RMSE With Stucture($\mu m$) |
|---|---|---|---|---|
| 5 | 0 | 0.04637 | 0.02143 | 0.001618 |
| 5 | 600 | 0.04637 | 0.02143 | 0.001618 |
| 5 | 1000 | 0.04629 | 0.02143 | 0.001618 |
| 10 | 0 | 0.09274 | 0.04285 | 0.003236 |
| 10 | 600 | 0.09289 | 0.04285 | 0.003236 |
| 10 | 1000 | 0.09061 | 0.04253 | 0.003229 |
| 20 | 0 | 0.1859 | 0.08391 | 0.00647 |
| 20 | 600 | 0.1825 | 0.08429 | 0.00645 |
| 20 | 1000 | 0.1799 | 0.09143 | 0.007279 |

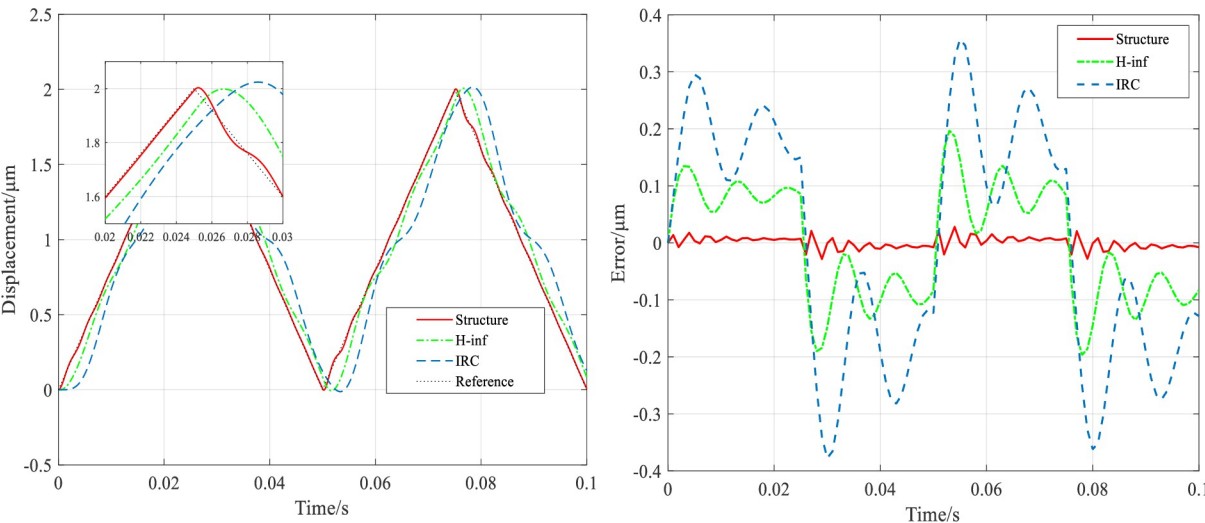

**Fig 11. Results of the system tracking a 20Hz grating scan signal at a mechanical load of 1000g.**

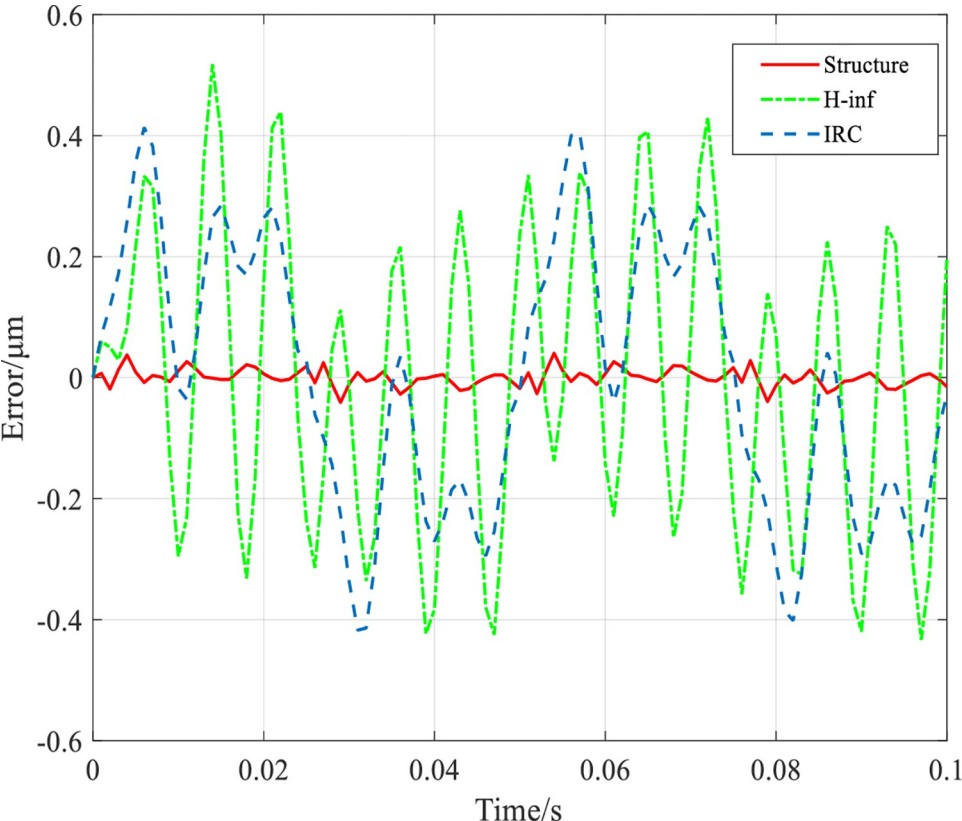

**Fig 12. The system tracks the 20Hz grating scan signal at a mechanical load of 1000g and a perturbation frequency of 139Hz.**

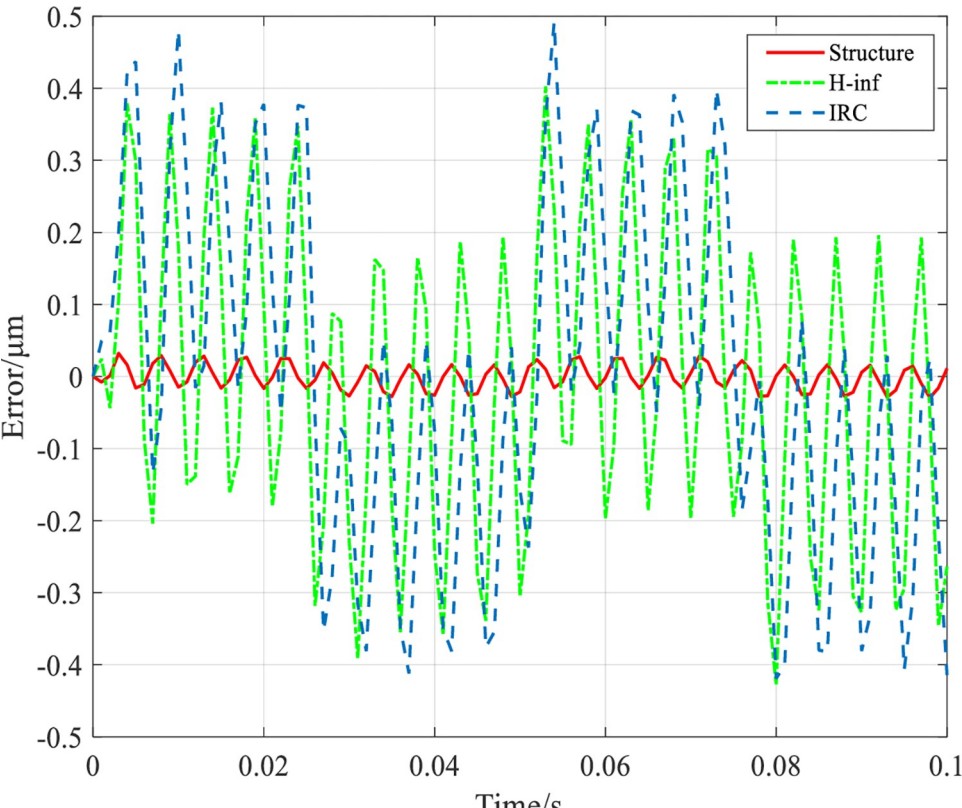

**Fig 13. The system tracks a 20Hz raster scan signal at a mechanical load of 0g and a disturbance frequency of 205Hz.**

## Conclusion

This paper deals with the combined problems of inherent resonant modes, low bandwidth and model uncertainty caused by mechanical load variation of the high-precision piezoelectric-driven nanopositioning platform. The system characteristics and performance requirements are analyzed, the structured $H_\infty$ control structure combining damping, robustness and tracking controller is designed, the appropriate weighting function is selected for each performance requirement, and finally the structured $H_\infty$ controller satisfying the comprehensive performance is given after parameter optimization. Realizing a piezoelectric-driven nanopositioning platform less than $0.007279\mu m$ (mechanical load tracking 5-20Hz grating signal at 0-1000g) with high accuracy, while also having a certain degree of robustness and anti-interference capability. Compared to IRC control, avoid the IRC controller parameter analysis design process does not take into account load changes and changes in the surrounding environment ignored robust stability problems, as well as improve the tracking accuracy. Compared to the traditional $H_\infty$ controller, it retains the comprehensive advantages of the $H_\infty$ design, avoids the high complexity of the $H_\infty$ controller and difficult to implement, multiple performance coupling constraints and design opacity problems, whose design structure is simple, low complexity and easy engineering implementation. Through experimental simulation verification, comparative analysis, the proposed structured $H_\infty$ controller has the comprehensive superiority of accuracy, response speed and robustness.

## Acknowledgments

Thanks to all the participants of the project team for their guidance in writing the paper.

## Author Contributions

**Conceptualization:** Huan Feng, Aiping Pang.

**Formal analysis:** Huan Feng, Congmei Jiang, Aiping Pang.

**Funding acquisition:** Congmei Jiang, Aiping Pang.

**Investigation:** Huan Feng, Hongbo Zhou.

**Methodology:** Huan Feng, Aiping Pang.

**Software:** Huan Feng.

**Supervision:** Congmei Jiang, Aiping Pang.

**Validation:** Huan Feng, Hongbo Zhou.

**Writing – original draft:** Huan Feng.

**Writing – review & editing:** Huan Feng, Hongbo Zhou, Congmei Jiang, Aiping Pang.

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
