## [Decision Letter · Decision Letter 0]

4 Jan 2023

PONE-D-22-33171High precision structured H∞ control of a piezoelectric nanopositioning platformPLOS ONE

Dear Dr. Pang,

Thank you for submitting your manuscript to PLOS ONE. After careful consideration, we feel that it has merit but does not fully meet PLOS ONE’s publication criteria as it currently stands. Therefore, we invite you to submit a revised version of the manuscript that addresses the points raised during the review process.

ACADEMIC EDITOR: Based on the received comments for the reviewers, this paper needs to be further revised. In my opinion, Major Revision is required.

We look forward to receiving your revised manuscript.

Kind regards,

He Chen

Academic Editor

PLOS ONE

Journal Requirements:

"This work was supported in part by the National Natural Science Foundation of China Regional Project 12162007, in part by the QianJiaoJi, China [2022]043 and in part by the QianKehe Foundation [2020] 1Y273."

"This work was supported in part by the National Natural Science Foundation of China Regional Project 12162007, in part by the QianJiaoJi, China [2022]043 and in part by the QianKehe Foundation [2020] 1Y273.

Sponsor: Ai-Ping Pang"

6. Please upload a copy of Figures 12 and 13, to which you refer in your text on pages 20 and 21. If the figure is no longer to be included as part of the submission please remove all reference to it within the text.

Additional Editor Comments:

Based on the received comments for the reviewers, this paper needs to be further revised. In my opinion, Major Revision is required.

Reviewers' comments:

Reviewer's Responses to Questions

**Comments to the Author**

1. Is the manuscript technically sound, and do the data support the conclusions?

Reviewer #1: Yes

Reviewer #2: Yes

2. Has the statistical analysis been performed appropriately and rigorously? 

Reviewer #1: Yes

Reviewer #2: Yes

3. Have the authors made all data underlying the findings in their manuscript fully available?

Reviewer #1: Yes

Reviewer #2: Yes

4. Is the manuscript presented in an intelligible fashion and written in standard English?

Reviewer #1: Yes

Reviewer #2: Yes

5. Review Comments to the Author

Reviewer #1: The paper focuses on the combined problems of inherent resonant modes, low bandwidth and model uncertainty of the piezoelectric-driven nanopositioning platform. The system characteristics and performance requirements are analyzed and the structured H-infinity controller is designed to satisfy the multiple performance requirements. Although the paper is interesting, the following issues should be addressed:

1. The references should be updated, the literatures are a bit old. Also, the references review in the introduction is incomplete, which should be further improved.

2. Since the paper first models the piezoelectric positioning system by considering the hysteresis effect, it may be better to discuss the classical hysteresis models, such as the Preisach model, Bouc-Wen model, Duhem model, Maxwell model, etc. In addition, some advanced neural network based hysteresis models should also be discussed, such as the gated recurrent unit based hysteresis model, improved neural Turing machine based hysteresis model, etc. The additional discussion can make readers understand the reason of adopting the modeling method in the paper.

3. Apart from the open-loop Bode diagram, it may be better to provide the input-output curves of the system to present the modeling performance more intuitively.

4. The author should pay attention to grammar mistakes, and the paper needs careful proof-reading.

Reviewer #2: This research is quite interesting by incorporating the newly proposed structured H-inf control into the nano-positioning control. As can be learnt, the order of the proposed controller is low for easy implementation. And another advantage is that the functions of each parts are explicable, which outperforms the conventional H-inf controller. Despite the advantages presented in this paper, there are some aspects that might help improve the quality and readability, listed as follows:

1. I'm curious about the high frequency interference signal, what's it? Sinusoidal signals or random signals?

2. For the choice of W2, eq(18), is it able to explain the way to choose W2? To be more specific, among several alternatives, how can one decide which is the best? What are the expected results on choosing different W2's? BTW: In fig5, zero-load bode diagram is expected as well.

3. In the introduction, when the structured H-inf integrated control is firstly mentioned at line 78, it should be more comprehensive to introduce the essence of the "structured H-inf control". Or in other words, briefly explain "what is a structured H-inf control".

4. The plots should be carefully redesigned, for example:

a) the text in the figure is small for printed versions.

b)the figures is not kind for color-blind people, and is not distinguishable in gray-printed version.

6. PLOS authors have the option to publish the peer review history of their article (what does this mean?). If published, this will include your full peer review and any attached files.

Reviewer #1: No

Reviewer #2: No

---

## [Author Response · Author response to Decision Letter 0]

15 Feb 2023

Response to editor:

1: This article has been modified to conform to PLOS ONE journal format.

2: The funding information has been deleted and modified in the Acknowledgements section.

3: Copies of Figures 12 and 13 have been uploaded, which are mentioned and described in the text, as shown in lines 419 and 422.

4:The data availability statement has been partially revised.

Response to reviewer 1:

Reviewer #1: The paper focuses on the combined problems of inherent resonant modes, low bandwidth and model uncertainty of the piezoelectric-driven nanopositioning platform. The system characteristics and performance requirements are analyzed and the structured H-infinity controller is designed to satisfy the multiple performance requirements. Although the paper is interesting, the following issues should be addressed:

1. The references should be updated, the literatures are a bit old. Also, the references review in the introduction is incomplete, which should be further improved.

Response: Some new references have been updated. In addition, to address the problem of incomplete review of references, this paper has been supplemented with a review of the feedforward/feedback combined control method, see lines 68-74 marked in red.

2. Since the paper first models the piezoelectric positioning system by considering the hysteresis effect, it may be better to discuss the classical hysteresis models, such as the Preisach model, Bouc-Wen model, Duhem model, Maxwell model, etc. In addition, some advanced neural network based hysteresis models should also be discussed, such as the gated recurrent unit based hysteresis model, improved neural Turing machine based hysteresis model, etc. The additional discussion can make readers understand the reason of adopting the modeling method in the paper.

Response: For classical hysteresis models such as physics-based models, phenomenological models including differential equation-based hysteresis models (common Duhem, Backlash-like and Bouc-Wen models), operator-based hysteresis models (typical Preisach models) and other hysteresis models (based on artificial neural networks, fuzzy systems and support vector machines, etc.) intelligent models), etc., which have been added to the discussion in the models section of this paper. See lines 164-181 marked in red.

3. Apart from the open-loop Bode diagram, it may be better to provide the input-output curves of the system to present the modeling performance more intuitively.

Response: The discussion in this paper focuses mainly on control methods, and due to the lack of a real experimental platform, its control object model is the model after system identification borrowed from reference [40], which has been marked in this paper with citations and specific details available in reference [40]. See lines 184-187 marked in red.

4. The author should pay attention to grammar mistakes, and the paper needs careful proof-reading.

Response: The paper has been corrected for multiple grammatical errors and has been carefully proofread.

Response to reviewer 2:

Reviewer #2: This research is quite interesting by incorporating the newly proposed structured H-inf control into the nano-positioning control. As can be learnt, the order of the proposed controller is low for easy implementation. And another advantage is that the functions of each parts are explicable, which outperforms the conventional H-inf controller. Despite the advantages presented in this paper, there are some aspects that might help improve the quality and readability, listed as follows:

1. I'm curious about the high frequency interference signal, what's it? Sinusoidal signals or random signals?

Response: External disturbance interference signal may be any form of high frequency signal. The high-frequency disturbance signal test subsection of this paper is selected as a sinusoidal signal close to the resonant frequency, respectively, and . The original text was omitted in the editing and has been revised, see lines 416-417 and lines 422-423 in red.

2. For the choice of W2, eq(18), is it able to explain the way to choose W2? To be more specific, among several alternatives, how can one decide which is the best? What are the expected results on choosing different W2's? BTW: In fig5, zero-load bode diagram is expected as well.

Response: (1): The original paper is indeed vague about the selection of W2, which has been explained and modified in the corresponding part of the paper, see lines 303-308 in red. The multiplicative uncertainty at 600g and 1000g load is calculated by equation (13), while for all the uncertainties, the appropriate weighting function W2 is designed. For all frequency bands, so that the multiplicative uncertainty needs to satisfy , where is the unitized regression, the selection of W2 can satisfy the above conditions.

(2): There is no best W2, and the expected result of choosing W2 is that the solved control system has robust stability, that is, the optimal parameters of the controller are obtained after satisfying , which is explained in the subsection on control performance analysis.

(3) Fig.5 shows the weighting function and the multiplicative regression diagram under load. The curves corresponding to 600 and 1000g are the multiplicative uncertainties under 600 and 1000g load calculated by equation (13), respectively, which is not a zero-load bode diagram and can be compared with Fig.2 or different. The original description of Fig. 5 is indeed misleading and has been revised.

3. In the introduction, when the structured H-inf integrated control is firstly mentioned at line 78, it should be more comprehensive to introduce the essence of the "structured H-inf control". Or in other words, briefly explain "what is a structured H-inf control".

Response: The essence of the structured H∞ control strategy has been added in that section. The core of structured H∞ control theory is a locally optimal control strategy that balances control performance and controller complexity. Firstly, the reasonable controller structure is designed according to the system characteristics and control objectives. Secondly, the "standard H∞ parametric matrix" with multiple performance decoupled outputs and multiple controllers mixed and nested is derived to build a standard structured integrated control form. Finally, the reasonable weighting function is designed to solve for the optimal controller parameters. See lines 84-90 marked in red.

4. The plots should be carefully redesigned, for example:

a) the text in the figure is small for printed versions.

b)the figures is not kind for color-blind people, and is not distinguishable in gray-printed version.

Response: a) The proposed figure has been modified for the part of the figure with smaller text, such as Figure 5, Figure 6, Figure 7, Figure 8, Figure 9 in (a), (b), (c) , Figure 11, Figure 12 and Figure 13.

b) In response to the data on the bar chart in Figure 10, which did appear to be unfriendly, the data have been removed and modified, and the values can be viewed in Table 1. Some color charts were used because the journal article was published in color upon review, and will be revised subsequently if there are any problems.

---

## [Decision Letter · Decision Letter 1]

29 Mar 2023

PONE-D-22-33171R1High precision structured H∞ control of a piezoelectric nanopositioning platformPLOS ONE

Dear Dr. Pang,

Thank you for submitting your manuscript to PLOS ONE. After careful consideration, we feel that it has merit but does not fully meet PLOS ONE’s publication criteria as it currently stands. Therefore, we invite you to submit a revised version of the manuscript that addresses the points raised during the review process.

We look forward to receiving your revised manuscript.

Kind regards,

He Chen

Academic Editor

PLOS ONE

Journal Requirements:

Additional Editor Comments:

Based on the received comments, there still exist some questions need further explanations. Minor Revision is required.

Reviewers' comments:

Reviewer's Responses to Questions

**Comments to the Author**

1. If the authors have adequately addressed your comments raised in a previous round of review and you feel that this manuscript is now acceptable for publication, you may indicate that here to bypass the “Comments to the Author” section, enter your conflict of interest statement in the “Confidential to Editor” section, and submit your "Accept" recommendation.

Reviewer #1: All comments have been addressed

Reviewer #2: All comments have been addressed

2. Is the manuscript technically sound, and do the data support the conclusions?

Reviewer #1: Yes

Reviewer #2: Yes

3. Has the statistical analysis been performed appropriately and rigorously? 

Reviewer #1: Yes

Reviewer #2: Yes

4. Have the authors made all data underlying the findings in their manuscript fully available?

Reviewer #1: Yes

Reviewer #2: Yes

5. Is the manuscript presented in an intelligible fashion and written in standard English?

Reviewer #1: Yes

Reviewer #2: Yes

6. Review Comments to the Author

Reviewer #1: (No Response)

Reviewer #2: 1. The place of the newly added content discussing the massive piezoelectric models is not suitable. It might be better to place it right at the beginning of the Section Model, rather than the middle.

2. Section name "Experimental comparative analysis" is misleading since there is no on-site experiments, e.g., only simulation is provided. It should be more informational.

3. Some more words on the plots:

a) For font size: The font size for the x/y-labels can be further enlarged. These label font size being much smaller than the normal font is not suggested unless there are adequate reasons. This rule applies to the legends as well.

b) For colors: Not all people prefer e-papers, some reads the printed version. So, it is suggested that the colors are chosen that they are distinguishable in gray-printed version. For example, distinguish them with different line styles (dashed, dotted, dash-dotted, solid). Proofreading the paper one a gray-printed one is a good habit.

c) For data values: In fig 10, the data values help comparison and should be kept. The modifications suggested are:

1) Adjust the text size;

2) Differ the bars with different patterns (different colors are welcomed at the same time);

3) Keep the data value at the top of the bars, in black.

7. PLOS authors have the option to publish the peer review history of their article (what does this mean?). If published, this will include your full peer review and any attached files.

Reviewer #1: No

Reviewer #2: No

---

## [Author Response · Author response to Decision Letter 1]

14 Apr 2023

Response to reviewer 2:

1. The place of the newly added content discussing the massive piezoelectric models is not suitable. It might be better to place it right at the beginning of the Section Model, rather than the middle.

Response: Added discussion of the model has been placed at the beginning of the model introduction

2. Section name "Experimental comparative analysis" is misleading since there is no on-site experiments, e.g., only simulation is provided. It should be more informational.

Response: The chapter title " Experimental Comparative Analysis" has been changed to "Simulation Comparative Analysis"

3. Some more words on the plots:

a) For font size: The font size for the x/y-labels can be further enlarged. These label font size being much smaller than the normal font is not suggested unless there are adequate reasons. This rule applies to the legends as well.

b) For colors: Not all people prefer e-papers, some reads the printed version. So, it is suggested that the colors are chosen that they are distinguishable in gray-printed version. For example, distinguish them with different line styles (dashed, dotted, dash-dotted, solid). Proofreading the paper one a gray-printed one is a good habit.

c) For data values: In fig 10, the data values help comparison and should be kept. The modifications suggested are:

1) Adjust the text size;

2) Differ the bars with different patterns (different colors are welcomed at the same time);

3) Keep the data value at the top of the bars, in black.

Response:

1) The text in the diagram has been resized.

2) The coloured lines have been distinguished by different line styles, which are also distinguishable in the grey printed version, see Figs. 2, 5, 6, 7, 8, 9, 11, 12 and 13 for the modified images.

3) Figure 10 has been modified to retain the data values at the top of the bars and the bars are differentiated by a different style.

---

## [Decision Letter · Decision Letter 2]

17 May 2023

High precision structured H∞ control of a piezoelectric nanopositioning platform

PONE-D-22-33171R2

Dear Dr. Pang,

We’re pleased to inform you that your manuscript has been judged scientifically suitable for publication and will be formally accepted for publication once it meets all outstanding technical requirements.

Kind regards,

He Chen

Academic Editor

PLOS ONE

Additional Editor Comments (optional):

Based on the received comments, this paper can be published now. Congratulations!

Reviewers' comments:

Reviewer's Responses to Questions

**Comments to the Author**

1. If the authors have adequately addressed your comments raised in a previous round of review and you feel that this manuscript is now acceptable for publication, you may indicate that here to bypass the “Comments to the Author” section, enter your conflict of interest statement in the “Confidential to Editor” section, and submit your "Accept" recommendation.

Reviewer #1: All comments have been addressed

Reviewer #2: All comments have been addressed

2. Is the manuscript technically sound, and do the data support the conclusions?

Reviewer #1: Yes

Reviewer #2: Yes

3. Has the statistical analysis been performed appropriately and rigorously? 

Reviewer #1: Yes

Reviewer #2: Yes

4. Have the authors made all data underlying the findings in their manuscript fully available?

Reviewer #1: Yes

Reviewer #2: Yes

5. Is the manuscript presented in an intelligible fashion and written in standard English?

Reviewer #1: Yes

Reviewer #2: Yes

6. Review Comments to the Author

Reviewer #1: The authors have addressed my comments. I have no more comments. The paper can be accepted for publication now.

Reviewer #2: All the comments are well-addressed, the data support the conclusions, therefore, acceptance is suggested.

7. PLOS authors have the option to publish the peer review history of their article (what does this mean?). If published, this will include your full peer review and any attached files.

Reviewer #1: No

Reviewer #2: No

---

## [Editor Report · Acceptance letter]

9 Jun 2023

PONE-D-22-33171R2 

High precision structured H∞ control of a piezoelectric nanopositioning platform 

Dear Dr. Pang:

I'm pleased to inform you that your manuscript has been deemed suitable for publication in PLOS ONE. Congratulations! Your manuscript is now with our production department. 

Kind regards, 

on behalf of

Dr. He Chen 

Academic Editor

PLOS ONE